# Significant reduction of PM2.5 in eastern China due to regional-scale emission control: Evidences from the SORPES station, 2011-2018

Aijun Ding[1,2], Xin Huang[1,2], Wei Nie[1,2], Xuguang Chi[1,2], Zheng Xu[1,2], Longfei Zheng[1,2], Zhengning Xu[1,2], Yuning Xie[1,2], Ximeng Qi[1,2], Yicheng Shen[1,2], Peng Sun[1,2], Jiaping Wang[1,2], Lei Wang[1,2], Jiannin Sun[1,2], Xiu-Qun Yang[1,2], Wei Qin[3], Xiangzhi Zhang[3,4], Wei Cheng[3], Weijing Liu[5], Liangbao Pan[4], and Congbin Fu[1,2]

[1]Joint International Research Laboratory of Atmospheric and Earth System Sciences, School of Atmospheric Sciences, Nanjing University, Nanjing, 210023, China

[2]Jiangsu Provincial Collaborative Innovation Center of Climate Change, Nanjing, China

[3]Jiangsu Environmental Monitoring Center, Department of Ecology and Environment of Jiangsu Province, Nanjing, China

[4]Department of Ecology and Environment of Jiangsu Province, Nanjing, China

[5]Jiangsu Provincial Academy of Environmental Science, Department of Ecology and Environment of Jiangsu Province, Nanjing, China

Correspondence to Aijun Ding (dingaj@nju.edu.cn)

**Abstract.** Haze pollution caused by PM2.5 is the largest air quality concern in China in recent years. Long-term measurements of PM2.5 and the precursors and chemical speciation is crucially important for evaluating the efficiency of emission control, understanding formation and transport of PM2.5 associated with the change of meteorology and for accessing the impact of human activities to regional climate change. Here we reported long-term continuous measurements of PM2.5, chemical components, and their precursors at a regional background station, the Station for Observing Regional Processes of the Earth System (SORPES), in Nanjing eastern China since 2011. We found that PM2.5 at the station has experienced a substantial decrease (-9.1%/yr), accompanied with even much significant reduction of SO2 (-16.7%/yr), since the national "Ten measures" for air took action in 2013. Control of open biomass burning and fossil-fuel combustion are the two dominant factors that influence the PM2.5 reduction in early summer and winter, respectively. In cold season (November-January), nitrate fraction was significantly increased, especially when air masses transport from north. More NH3 available from a substantial reduction of SO2 and increased oxidization capacity are the main factors for the enhanced nitrate formation. The changes of year-to-year meteorology contributed to 24% of the PM2.5 decrease since 2013. This study highlights several important implications on air pollution control policy in China.

# 1 Introduction

Fine particulate matter, with aerodynamic diameter smaller than 2.5 µm (PM$_{2.5}$), which impacts negatively on human health and visibility (Cao et al., 2012; Zhang et al., 2017; Malm et al., 2004), has been considered as one of the main air pollutants in China (He et al., 2001; Yao et al., 2002; Sun et al., 2006; Pathak et al., 2009; Zhang et al., 2015; Wang et al., 2017). To tackle this great challenge, China recently has implemented the Action Plan for Air Pollution Prevention and Control (i.e. the so-called national "Ten Measures" for air) in several developed critical regions since 2013 (Sheehan et al., 2014; Wang et al., 2017; Liu et al., 2018; Zheng et al., 2018; Liu et al., 2019). Measurement data, mainly from the ambient air quality monitoring network, showed some evidences of improved haze pollution in many cities in recent years (Wang et al., 2017; Lang et al., 2017; Zhang et al., 2019). However, to give a robust and quantitative assessment of the change due to specific emission reduction, high quality long-term continuous measurements of PM$_{2.5}$ and its chemical composition and precursors together with comprehensive data analysis and model simulations are needed because PM$_{2.5}$ has complex chemical compositions, and sources and formation mechanisms (Pathak et al., 2009; G. Wang et al., 2016; Cheng et al., 2016; Wen et al., 2018), and strong dependence on year-to-year meteorology (Zheng et al., 2015; Zhang et al., 2016; Zhang et al., 2019).

The Yangtze River Delta (YRD) is one of the developed and highly populated regions in China (Ding et al., 2013a; H. Wang et al., 2016). Under unfavorable meteorological conditions, PM$_{2.5}$ in this region could reach to very high concentration in winter and early summer, contributed mainly by fossil fuel combustion and agricultural straw burning, respectively (Ding et al., 2013ab; Cheng et al., 2014; Huang et al., 2016; Ding et al., 2016a; H. Wang et al., 2016; Wang et al., 2018). Some recent studies reported the change of PM$_{2.5}$ and its chemical components in some YRD cities in a short period, e.g. 2-3 years (Wang H. et al., 2016; Wang et al., 2017; Sun et al., 2018), however, so far there is a lack of long-term observational study with comprehensive measurements covering the entire period of national "Ten measures" period, i.e. 2013-2017, in this region.

In this study, we report the long-term continuous ground-based measurements of PM$_{2.5}$ and its chemical compositions as well as main precursors at the Station for Observing Regional Processes of the Earth System (SORPES) in Nanjing, western YRD for the period of 2011-2018. Based on Lagrangian dispersion modeling and comprehensive analysis with other supporting data, we investigate the impacts of emissions from fossil fuel combustion and open biomass burning (BB) and of year-to-year meteorology on the trend of primary and secondary PM$_{2.5}$ in this region.

## 2 Data and Methods

### 2.1 Brief introduction to the SORPES station and instrumentations

The SORPES station (118°57′10" E, 32°07′14" N, 62 m a.s.l.) is a cross-disciplinary research and experiment platform established in 2011 to understand the impact of human activities in the rapidly urbanized and industrialized eastern China region (Ding et al., 2013a; 2016b). Because of the unique geographical location, i.e. downwind of the North China Plain (NCP) and the YRD city cluster but upwind of Nanjing downtown (with distance about 20 km), this site can be considered as a regional background station for air quality studies in eastern China (Ding et al. 2016b). Regional anthropogenic plumes from the NCP to YRD city clusters and the early-summer open BB smoke in eastern China (Fig. 1) can influence this site under a complex multi-scale transport associated with Asian monsoon (Ding et al., 2013a; Ding et al. 2016b).

Continuous measurement of $PM_{2.5}$ mass concentration and its precursors, e.g. sulfur dioxide ($SO_2$) and nitrogen oxides ($NO_x$) (nitric oxide (NO) + nitrogen dioxide ($NO_2$)) etc., started in August 2011. More species, such as $PM_{2.5}$ chemical compositions, including black carbon (BC) and water-soluble ions (e.g. sulfate ($SO_4^{2-}$), nitrate ($NO_3^-$), ammonium ($NH_4^+$), potassium($K^+$), and calcium ($Ca^{2-}$), etc.) has been measured since 2013 (Xie et al., 2015; Sun et al., 2018; Wang et al., 2018; Shen et al., 2018). Details of the data, including instrumentation, measurement period, and data coverage, used in this study are given in Table S1 in the Supporting Information (SI). Briefly, instruments/analyzers for chemical compositions measurement are housed in a 2-floor building on the top of a small hill about 42 m above ground level. $PM_{2.5}$ mass concentration is measured by the online analyzer based on the light scattering and beta ray absorption method (Thermo Fisher Scientific, 5030SHARP, USA). The water-soluble inorganic ions, including $SO_4^{2-}$, $NO_3^-$, $NH_4^+$, and $K^+$ etc., are detected by the instrument for Measuring AeRosols and GAses (MARGA, Metrohm, Switzerland) (Xie et al, 2015; Sun et al, 2018). BC was measured using a 7-wavelenth Aethalometer (AE-31, Magee Scientific) and the data at a wavelength of 880 nm was used in this study (Shen et al., 2018; Virkkula et al., 2015). $SO_2$ and $NO_x$ are measured by the online analyzers with a time resolution of 1–5 min (Thermo Fisher Scientific, 43i and 42i, respectively). All the instruments are routinely calibrated for different durations. During the continuous measurement period, most of the instruments have a data coverage over 80% (Table S1).

### 2.2 Lagrangian dispersion modeling and other data sources

In order to help quantitatively understand the influence of year-to-year change in meteorology on air pollutant transport and dispersion, we conducted backward Lagrangian Particle Dispersion Modeling (LPDM) using the Hybrid Single-Particle Lagrangian Integrated Trajectory

(HYSPLIT) model (Stein et al., 2015). We used the model to calculate single-particle backward trajectories and to conduct cluster analysis. We also estimated the hourly $PM_{2.5}$ concentration based on the particle dispersion simulations following a method developed by Ding et al. (2013a; 2013c). Briefly, for each hour during the study period, the model was run 2 days and 7 days

backwardly with 3,000 particles released every hour at the altitude of 100 m over the SORPES station. The model calculated the position of particles by mean wind and a turbulence transport component, and the spatiotemporal distribution of these particles were further used to calculate the potential source contribution by using footprint "retroplume" (i.e., the residence time at the altitude of 100 m above the ground surface) and an emission inventory (Ding et al., 2013a;

2013c). Global Data Assimilation System (GDAS) data were used to drive the model and the MIX emission inventory database for the year of 2010 (Fig. 1a) (M. Li et al., 2017) was used for a quantitative estimation of $PM_{2.5}$ concentration.

Besides the observations at the SORPES station and data used in the LPDM simulations, various data are used to support the data analysis and discussions. To identify the impact of

agricultural straw burning, we used BB emission inventory from the Global Fire Emissions Database, Version 4.1 (with small fires) (GFED4s) (Giglio et al., 2013) and MODIS (Moderate Resolution Imaging Spectroradiometer) Thermal Anomalies/Fire daily L3 Global Product (MOD/MYD14A1) (Boschetti et al., 2009). The ERA5 reanalysis data (https://software.ecmwf.int/wiki/display/CKB/ERA5+data+documentation) from European

Centre for Medium-Range Weather Forecasts (ECMWF) together with the Tropical Rainfall Measuring Mission (TRMM) satellite observed precipitation (Huffman et al., 2007) are also use to investigate the year-to-year difference of meteorology that may influence the $PM_{2.5}$ concentration.

**3 Results and Discussion**

Based on continuous measurement at the SORPES station, Fig. 2 shows the trends of $PM_{2.5}$ mass concentration and key precursors ($SO_2$ and $NO_x$ ($NO+NO_2$)) since 2011, and the main $PM_{2.5}$ chemical components (BC, $SO_4^{2-}$, $NO_3^-$, and $NH_4^+$) since 2013. Considering the difference in the observation duration and the specific emission control in east China associated with the national "Ten Measures" for air since 2013 (Sheehan et al., 2014; Wang et al., 2017; Liu et al., 2018), we

conducted linear regression for the two periods: August 2011-July 2018 and August 2013- July 2018, respectively. It can be found that $PM_{2.5}$ concentration and the mixing ratio of two precursors show an overall decreasing trend during the past seven years (-6.4%/yr, -12.1%/yr, -4.6%/yr, and -11.1%/yr for $PM_{2.5}$, $SO_2$, $NO_2$ and $NO$, respectively), but more remarkable decreasing trends (-9.1%/yr, -16.7%/yr, -5.2%/yr, and -14.1%/yr for $PM_{2.5}$, $SO_2$, $NO_2$ and $NO$,

respectively) since 2013. For $SO_2$, the 5-yr reduction almost reached 70-80% and showed

significantly higher reduction rate in comparison with NOx. It demonstrates that the YRD region, as one of the main industry bases with a huge consumption of coal, achieved a very big success of air pollution prevention from desulfurization in power plants and factories and from replacement of coal with natural gases and electricity in recent years. In fact, a national wide significant reduction of $SO_2$ in the past few years has been also reported by ground and satellite measurements and emission estimations (C. Li et al., 2017; Liu et al., 2018; Zheng et al., 2018).

Fig. 2d shows that $NO_2$, another precursor of nitrate, shows a decreasing trend (-5.2% /yr) since 2013, which is much smaller than that of $SO_2$. Accordingly, the two secondary inorganic water-soluble ions, $SO_4^{2-}$ and $NO_3^-$ showed different trends, with the former a more significant reduction (-10.6%/yr vs. -5.8% /yr). For BC, an important particle mainly from primary emission of incomplete combustion but with significant impact to climate and aerosol-boundary layer feedback (Bond et al., 2013; Ding et al., 2016a; Wang et al., 2018; Huang et al., 2018), it showed a decreasing trend (about -8.4%) between those of sulfate and nitrate. Although BC is a short-lived climate forcer contributing to global warming (Bond et al., 2013; IPCC, 2013; Ding et al., 2016a), so far there are no specific reduction policy focus on it. Here the results show that the efforts in reducing $PM_{2.5}$ also caused BC reduction, which co-benefited to the mitigation of global warming (Bond et al. 2013, IPCC, 2013).

In eastern China, agricultural straw burning is particularly strong in early summer, i.e. from middle-May to middle-June, after the harvest of wheat (Ding et al., 2013ab; Cheng 2013; Huang et al., 2016; Chen et al., 2017). Intensive emission from these activities could cause a second maximum of $PM_{2.5}$ in early summer (Ding et al., 2013a), and also severe haze events with high concentration of $PM_{2.5}$ and other pollutants. For example, the SORPES station recorded an hourly concentration of $PM_{2.5}$ over 400 µg m$^{-3}$ in 10 June 2012 (Ding et al., 2013a; Xie et al., 2015; Nie et al., 2015). To examine the change before and after the "Ten measures" took action, Fig. 3a presents the seasonal variation of $PM_{2.5}$ mass concentration averaged for the periods of 2011-2014 and 2015-2018, respectively. It clearly shows that the secondary maximum of $PM_{2.5}$ mass concentration in early summer was missing in recent years, instead of a relative flat change (Fig. 3a).

Since the "Ten measures" took action in 2013, Chinese government has conducted very strict emission control of agricultural straw burning by using real-time satellite as a tool to monitor these activities (Chen et al., 2017; Wang et al., 2017). The MODIS satellite fire counts data did demonstrate the significant reduction in BB activities in this region since 2013 (Fig. 3b). To further investigate the potential impacts of other factors, i.e. the change of air pollutant transport associated with circulation patterns, we conducted LPDM simulations for the BB seasons (15 May – 20 June) during the two three-year periods. Fig. 4 shows that the averaged transport

pathways of air masses didn't change much for the two periods, but the MODIS satellite fire counts showed a distinct difference in both total amount and spatial distribution. These BB emitted smoke could be transported to the SORPES station in two days when the north wind prevailed.

Fig. 3b also shows the averaged TRMM precipitation in the same area as the fire counts data (i.e. dashed square in Fig. 1b) during 15 May-20 June, 2011-2018. It is indicative of a certain reverse correlation of precipitation with the total amount fire counts. However, the relationship between precipitation and BB is complicated. On one hand, the harvest season is generally before the "Meiyu" season in this region (Kitoh & Takao, 2006). Farmers usually choose continuous sunny day to harvest and to dry the wheat, while the straws are also easy to be burned in a dry condition (Feng et al., 2019). On the other hand, precipitation can influence wet deposition of smoke (Uematsu et al., 2010). Meanwhile, BB smoke has been demonstrated to modify rainfall. For example, Ding et al. (2013b) and Huang et al. (2016) reported a case of suppressed rainfall and changed rainfall pattern by BB smoke in this region based on the SORPES observations and numerical modeling. Although more quantitative studies are still needed, here the year-to-year variation of precipitation shouldn't be the dominant factor influencing the substantial decreasing the $PM_{2.5}$, especially for the period between 2013-2018 when the precipitation didn't show a significant trend.

To confirm the impact of BB on the $PM_{2.5}$ reduction, we further investigate the measured BB tracer, fine particulate $K^+$, as previous studies (Ding et al., 2013b; Xie et al., 2015; Nie et al., 2015; Zhou et al., 2017) suggested that it is a good tracer for agricultural straw burning in this region. The scatter plot of $K^+$ as a function of $PM_{2.5}$ mass concentration, color-coded with time, given in Fig. 3b clearly shows a dramatic decrease in the $K^+/PM_{2.5}$ proportion at the SORPES station, especially after 2015. Using the concentration of $K^+$ as a threshold, we identified BB ($K^+$ higher than the 75% percentile) and non_BB cases ($K^+$ lower than 25% percentile) and showed the scattering plots of sulfate and nitrate as a function of $PM_{2.5}$ in Fig. 5. It confirms that very efficient control of open BB emission is the dominant factor that influenced the early-summer $PM_{2.5}$ reduction in this region.

Besides the early summer, months in the cold season, e.g. November, December and January (NDJ), experienced strong $PM_{2.5}$ reduction in the past few years (Fig. 3a). In order to further explore the inter-relationship among precursors and secondary and primary particles, we show the scatter plots of different species color-coded with time in Fig. 6. For the $SO_2$-$NO_x$ plot, it changed from a bifurcation pattern in the earlier years to a latest monotonous one (Fig. 6a). The bifurcated pattern of $SO_2$ versus $NO_x$ indicates the impact of emissions from elevated coal burning point sources (with high $SO_2/NO_x$ ratio) and scattered vehicle sources (Wang et al.,

2002). The pattern change in recent years further confirms that the reduction of $SO_2$ was mainly due to efficient control from large elevated coal burning sources, such as power plants, and coal replacement with natural gas or electricity (Wang et al., 2002. Liu et al., 2018).

For the scatter plot of SNA versus BC (Fig. 6b), a less-changed SNA/BC slope from 2014 to 2018 somehow suggests the co-benefited reduction of short-lived climate forcers from mitigation of haze pollution in China. Since SNA are mainly secondary products due to oxidation of $SO_2$ and $NO_x$ and neutralization of $NH_3$ (Pinder et al., 2007) and BC is a tracer of primary pollutants from combustion sources (Bond et al., 2013), the similar slope here also indicates that the overall proportion of secondary particles in $PM_{2.5}$ was less changed. However, the sulfate/$PM_{2.5}$ ratio was significantly reduced from 2014 to 2018 (Fig. 6c), linked to the remarkable reduction of its precursor $SO_2$ as shown in Fig. 2b.  While for nitrate, an overall increased nitrate/$PM_{2.5}$ ratio could be clearly seen in recent years, especially for 2018, despite of a moderate decreasing of $NO_2$ (Fig. 2d). It has been well known that $PM_{2.5}$ has a nonlinear response to the reduction of sulfate because decreases in sulfate may increase aerosols when more nitric acid may enter the aerosol phase (West et al., 1999; Pinder et al., 2007; Liu et al., 2019). Here the regional scale sulfate reduction should have increased the availability of $NH_3$ for the formation of nitrate (Pathak et al., 2009; Huang et al., 2012; Wang et al., 2011; Liu et al., 2018; Liu et al., 2019). In addition, the increased atmospheric oxidization capacity will also enhance the formation of nitrate (Wen et al., 2018; Liu et al., 2019).

To further investigate the change in SNA and BC associated with air masses from main source regions in eastern China, we carried out back-trajectory cluster analysis for the months of NDJ during 2013-2018 (Fig. S1). In Fig. 7, we show the pie charts of SNA and BC in $PM_{2.5}$ at SORPES between 2013 and 2017, i.e. the beginning and end of the first 5-year "Ten measures", for air masses originated from NCP (Figs. 7a,d), Central-Eastern China plain (Figs. 7b,e), and the YRD  (Figs. 7c,f), respectively. The results show a significantly higher reduction (-52.3%) in the air masses from NCP than the other two (~32%), indicating more strict and efficient control in North China (Liu et al., 2019; Li et al., 2019). For the fraction of chemical composition, sulfate show significant decrease of from NCP (from 23% to 19%) and YRD (from 18% to 14%), but nitrate show significant increase (from 24% to 29% for YRD air masses and to 32% for air masses from the two regions in the north). While ammonium shows 1% increase in all the three regions. These results confirmed the increased availability of $NH_3$ for the formation of SNA (Pathak et al., 2009; Liu et al., 2018, Liu et al., 2019; Li et al., 2019), and suggest a stricter $NH_3$ emission reduction as a potentially efficient means for further $PM_{2.5}$ mitigation in the eastern China.

Meteorological conditions, especially transport and dispersion, could significantly influence PM$_{2.5}$ concentration (Ding et al., 2013ab; Zhang et al., 2016). From the air quality management perspectives, it is very important to quantify the influence of emission reduction and year-to-year change in meteorology based on the observed trend of air pollutants. Because NDJ are the three months with highest PM$_{2.5}$ concentration at SORPES (Fig. 3a), the trend during these months should also dominate the overall trend in annual average. In addition, the total precipitation during this period were relatively low (upper panel of Fig. 7), indicating a limited influence of wet deposition. Considering the limitation of LPDM in characterizing wet deposition and also secondary PM formation, we only chose NDJ to conduct the LPDM simulations based on the fixed MIX emission inventory for 2010 (M. Li et al., 2017) to quantify the impacts from emission reduction and from the change in meteorology. To better characterize the influence of year-to-year differences in meteorology alone based on LPDM, we scaled the simulation results to make the modeled median values equal to observation for the first year, i.e. November 2011-January 2012. This procedure removes other systematic differences between the LPDM simulation and the observation, e.g. uncertainty of emission inventory, parameterization of boundary layer etc. in the model. In fact, for the results with all years scaled using this method, the LPDM simulations could well reproduce the day-to-day variation of PM$_{2.5}$ (Fig. S2). The good agreement implies that the primary and secondary PM$_{2.5}$ had a consistent change with day-to-day weather, which controls the transport and dispersion of pollutants.

Fig. 8 shows the trends of observed PM$_{2.5}$ concentration at the SORPES station and the scaled LPDM simulation results to the first-year observation (i.e. 2011/2012) in NDJ from 2012 to 2018. The observations in the three months show a consistent trend with the annual average given in Fig. 1a. From the LPDM simulation results based on fixed emission inventory, we can find that the meteorology-influenced PM$_{2.5}$ had strong year-to-year variation, with a minimum medium value as 84.8 µg m$^{-3}$ in 2011/2012 to 108.4 µg m$^{-3}$ in 2013/2014, i.e. with a year-to-year difference up to 28%. The difference in the two years were mainly due to different transport and dispersion patterns related to different large-scale circulations (Fig. 9). In the NDJ of 2013/2014, the eastern China were dominated by a stagnant High isolated from the continental High, causing more sub-regional influence of air masses in the eastern China (Fig. 9d) than the NDJ of 2011/2012. For the LPDM simulations with fixed emission inventory, an increasing trend (about 1.11 %/yr) existed for the entire 7 years but a moderate decreasing trend (-2.6%/yr) in the latest five years (Fig. 8). Here the distinctly different trends for the 5-year and 7-year periods indicate that the interannual variation could substantially influence the understanding of trend in air quality for a period less than one decade. For the period since 2013, i.e. when the "Ten measures" took action, the observed decrease of PM$_{2.5}$ (-10.9%/yr) are mainly induced by emission reduction (76%) and the year-to-year change in mereology contributed to the left 24%,

i.e. about a quarter of the overall one. Here our estimation is consistent with that by Zhang et al. (2019), in which meteorology was estimated to contribute 20%-30% of the deceased trend in the YRD region during the period of 2013-2017.

**4 Conclusions**

In this study, we make a comprehensive analysis based on long-term continuous high-quality measurements of $PM_{2.5}$, relevant chemical compositions and precursors at a regional background station SORPES in Nanjing, eastern China since 2011. By utilizing Lagrangian dispersion modeling and various data, we quantified the impact of changes in different emission sources

and in year-to-year meteorology on the observed trends, especially after the implementation of Action Plan for Air Pollution Prevention and Control (i.e. the national "Ten measures" for air) in 2013. Main conclusions are given below:

1)   We found a substantial reduction of $PM_{2.5}$ together with stronger reduction of $SO_2$ at the SORPES station during 2011-2018, and much faster decreasing trends (-9.1%/yr in $PM_{2.5}$

and -16.7%/yr in $SO_2$) associated with concurrent reduction in sulfate (-10.6%/yr), nitrate (-5.8%/yr) and BC (-8.4%/yr) during 2013-2018, i.e. after the "Ten measures" implemented.

2)   The early-summer agricultural straw burning has been found to be significantly reduced since 2013 in the eastern China. It significantly reduced $PM_{2.5}$ concentration at the

SORPES station during May-June, resulting in a change in the seasonal pattern of $PM_{2.5}$. In cold season (November-January), the fraction of nitrate in $PM_{2.5}$ was significantly increased, especially when air masses came from north. Besides an increased oxidization capacity, more $NH_3$ were available for nitrate formation in the condition of reduced sulfate associated with a substantial reduction of $SO_2$ are the main reason causing the

enhanced nitrate formation.  A stricter $NH_3$ emission reduction is suggested for the further $PM_{2.5}$ mitigation in the eastern China.

3)   The year-to-year difference in meteorological conditions could cause strong change in wintertime $PM_{2.5}$ concentrations by mainly influencing the transport and dispersion of air pollutants. The change in meteorology contributed to 24% of the observed decrease in

$PM_{2.5}$ at the SORPES station in November-January during 2013-2018, with the left 76% caused mainly by emission reduction.

This study demonstrates the unique role of long-term high-quality continuous measurement at a regional background station in the understanding of the impact of emission sources, chemical mechanisms and meteorology processes on the change of atmospheric components.

Based on comprehensive and in-depth analysis of the data, the long-term measurement can provide a quantitative understanding about the large-scale air quality measures and can also raise some insights on the policy making for future air pollution prevention and control.

5 **Author contributions.** AD designed this study, carried out the data analysis, and wrote the paper with contributions from all co-authors. XH participated the data analysis, modeling and plotted some figures. WN, XC, ZX, LZ, ZNX, YX, XQ, YS, PS, JW, LW, WQ and XZ conducted the measurements at the SORPES station.

**Competing interests.** The authors declare that they have no conflict of interest.

**Acknowledgments**

This work was funded by the Ministry of Science and Technology of the People's Republic of China (2016YFC0200500; 2018YFC0213800; 2016YFC0202000) and the National Natural Science Foundation of China (41725020 and 41621005). We thank colleagues and students in the School of Atmospheric Sciences at Nanjing
15 University and Prof. Markku Kulmala's group at University of Helsinki for their contributions on the development of SORPES station and the maintenance of the measurements. The ERA-5 global reanalysis data are available at http://apps.ecmwf.int/datasets/data. MIX emission data are available at http://www.meicmodel.org/dataset-mix.html. Biomass emission data are available at https://daac.ornl.gov/cgi-bin/dsviewer.pl?ds_id=1293. MODIS satellite fire counts data are available at https://e4ftl01.cr.usgs.gov/MOTA/ and the TRMM precipitation data are available at
20 http://disc.gsfc.nasa.gov/datacollection/3B42.V7.html. The SORPES data in this paper will be publicly available as the paper published.

**Supporting Information**

Figs. S1-2 and Table S1

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

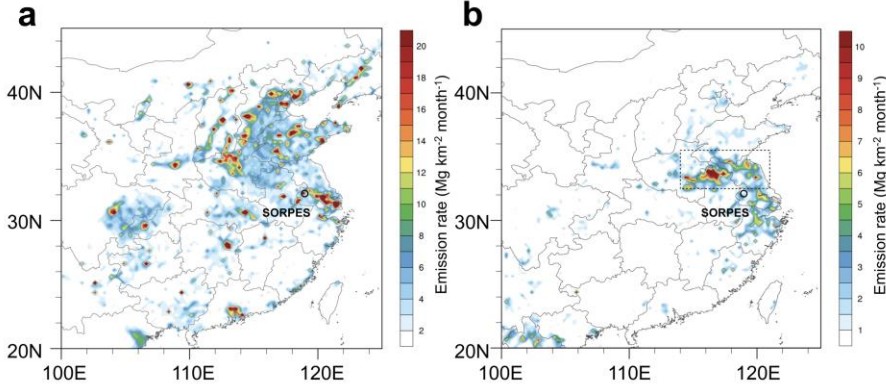

**Figure 1.** Spatial distributions of **(a)** anthropogenic emission of primary PM$_{2.5}$ in 2010 (Data from the MIX inventory available in MEIC database of Tsinghua University) and **(b)** averaged carbon emission from open biomass burning in May-June during 2012-2017 (Data from GFEDv4 available at https://daac.ornl.gov/cgi-bin/dsviewer.pl?ds_id=1293).

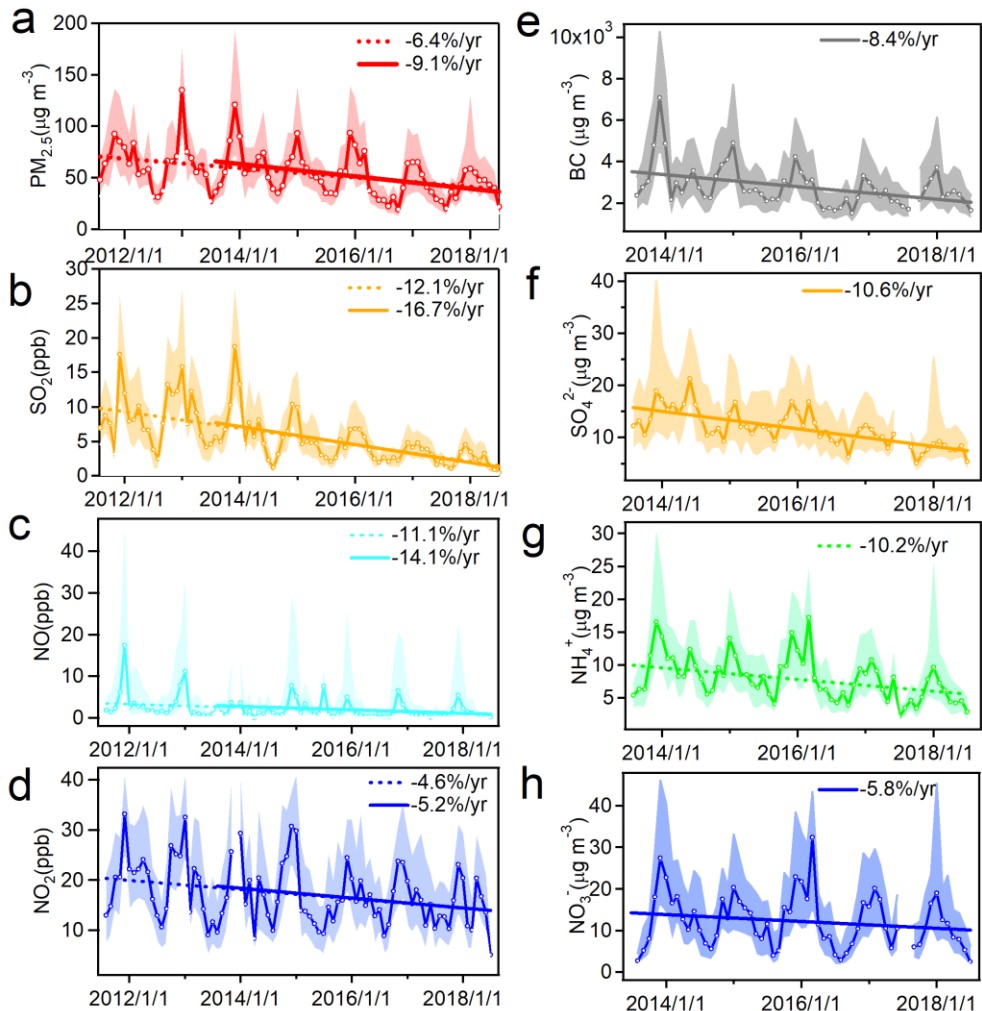

**Figure 2.** Monthly statistics and trends for **(a)** PM$_{2.5}$, **(b)** SO$_2$, **(c)** NO, **(d)** NO$_2$, **(e)** BC, **(f)** SO$_4^{2-}$, **(g)** NH$_4^+$, and **(h)** NO$_3^-$ observed at the SORPES station. Note: For BC, SO$_4^{2-}$, NH$_4^+$ and NO$_3^-$, only data during August 2013-July 2018 are shown here. The solid lines marked with open cycles represent the monthly medium value and shaded areas mark the data from 25$^{th}$ to 75$^{th}$ percentiles. Dashed and solid lines show the linear regression fitting for data during 2011-2018 and 2013-2018, respectively.

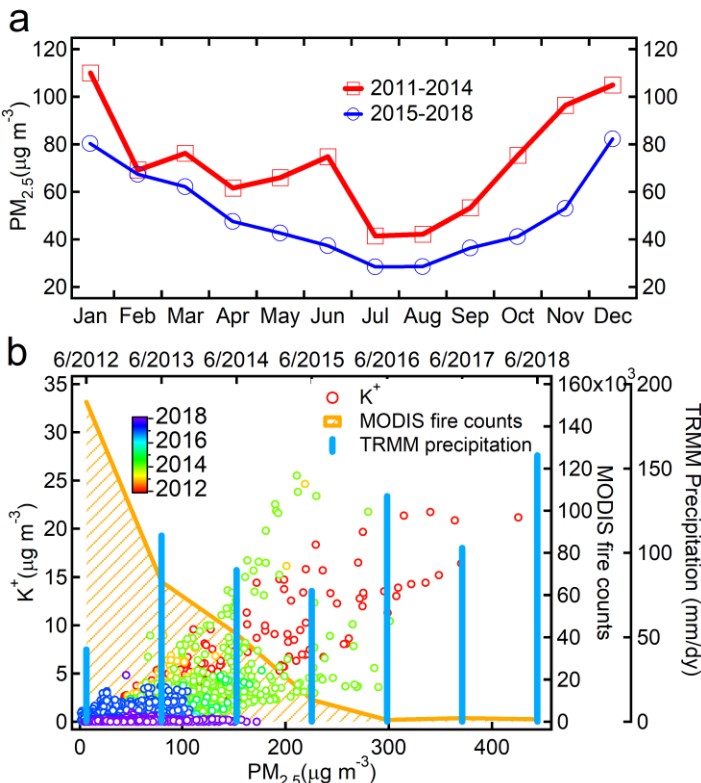

**Figure 3.** **(a)** Seasonal variation of PM$_{2.5}$ mass concentration measured at the SORPES station during 2011-2014 and 2015-2018, and **(b)** Scatter plots of K$^+$ as a function of PM$_{2.5}$ concentration measured at the SORPES station and the sum of MODIS fire counts and of TRMM precipitation in the BB domain (square in Fig. 1b) during 15 May-20 June, 2012-2018.

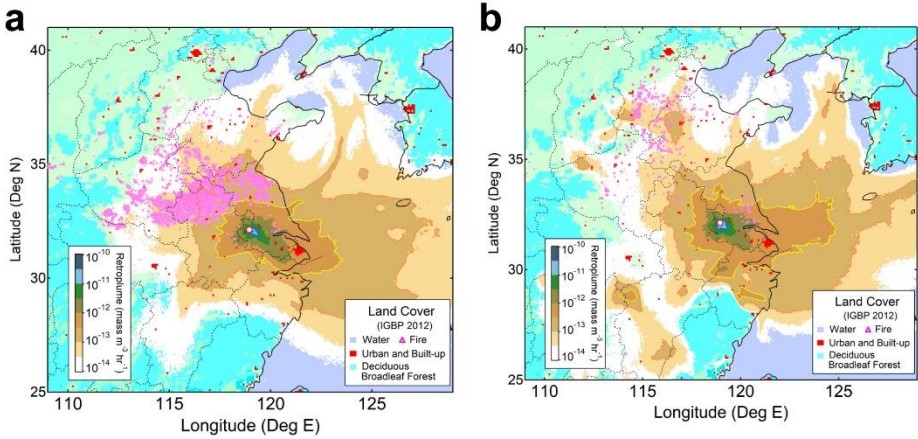

**Figure 4.** Averaged retroplume from 2-day backward Lagrangian dispersion modeling and MODIS fire counts for the period of 15 May- 20 June during **(a)** 2012-2014 and **(b)** 2016-2018.

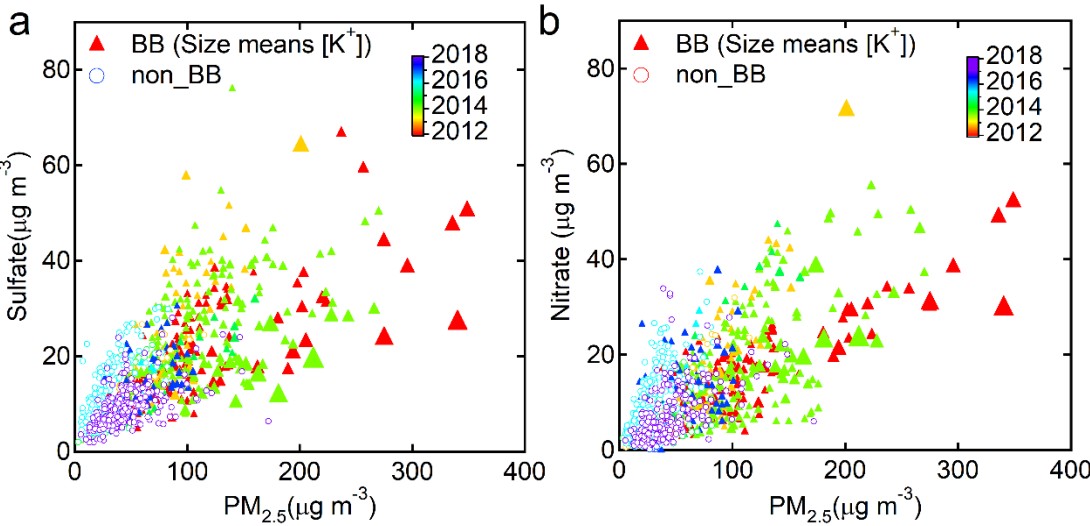

**Figure 5.** Scatter plot of **(a)** sulfate and **(b)** nitrate as a function of PM$_{2.5}$ for biomass burning (with K$^+$ higher than 75% percentile) and non-biomass burning (with K$^+$ lower than 25% percentile) cases for the period of 15 May- 20 June during 2012-2018.

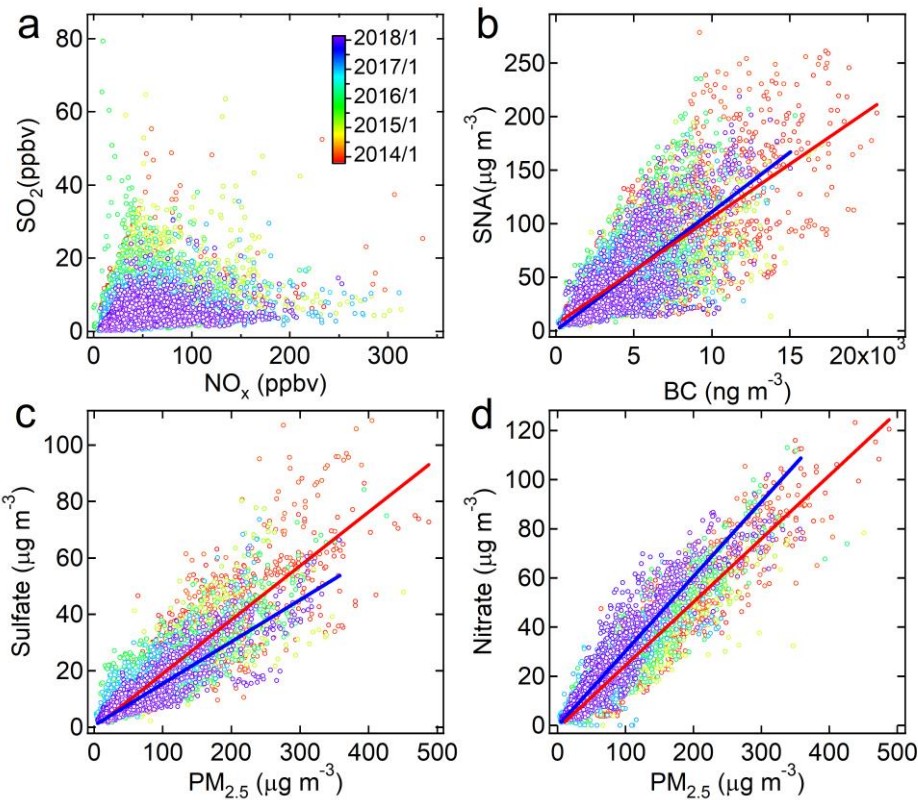

**Figure 6.** Scatter plots of **(a)** $SO_2$ versus $NO_x$, **(b)** SNA(i.e. the sum of sulfate, nitrate and ammonium versus BC), **(c)** $SO_4^{2-}$ versus $PM_{2.5}$, and **(d)** $NO_3^-$ versus $PM_{2.5}$ in November, December and January during 2013-2018.

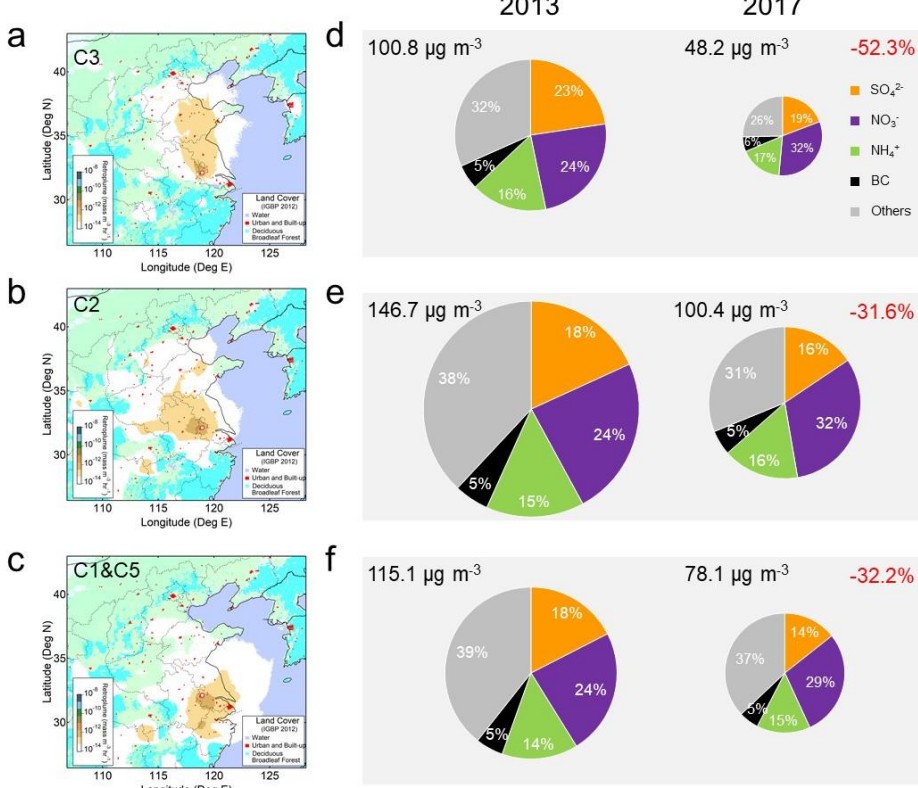

**Figure 7.** Averaged retroplume for air masses from **(a)** NCP (C3 in Fig. S1), **(b)** Central-Eastern Plain (C2 in Fig. S1), and **(c)** YRD (C1 & C5 in Fig.S1),  and **(d-f)** a comparison of pie charts of SNA and BC in PM$_{2.5}$ for the three types of air masses in NDJ of 2013 (November and December 2013 and January 2014) and 2017 (November and December 2017 and January 2018), corresponding to the three transport pathways of different air masses given in (a-c). Note: Averaged PM$_{2.5}$ concentrations for the corresponding air masses and periods are shown in the top-left of each pie chart and also indicated by the size of the pie charts. The reduction rates between the two periods are shown in red color in the top left of right panel (in percentage).

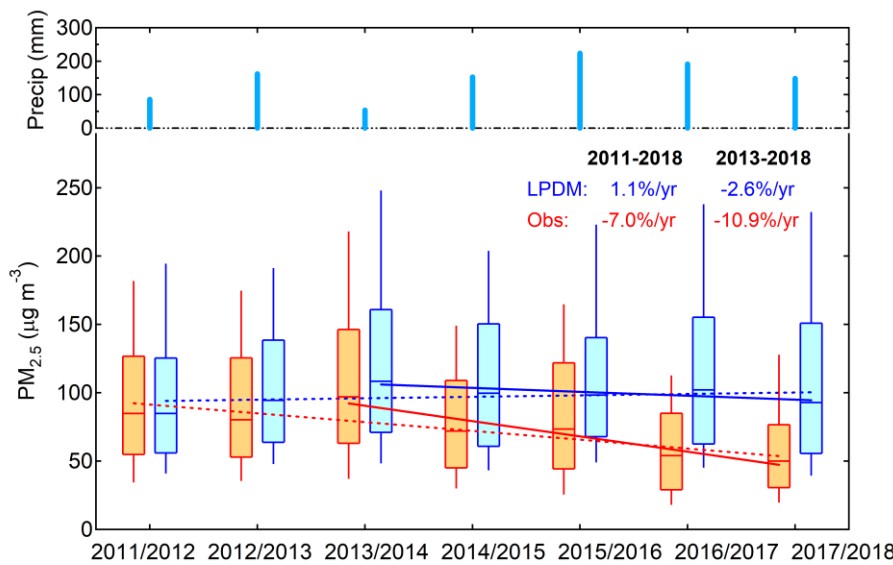

**Figure 8.** Statistics of PM$_{2.5}$ concentrations from the SORPES observation and the scaled LPDM simulations to the first-year observation (lower panel) and a sum of 3-month TRMM precipitation averaged in a 2°×2° grid (118° E-120° E, 31° N -33° N) around the SORPES station (upper panel) for November-December-January during 2011-2018.

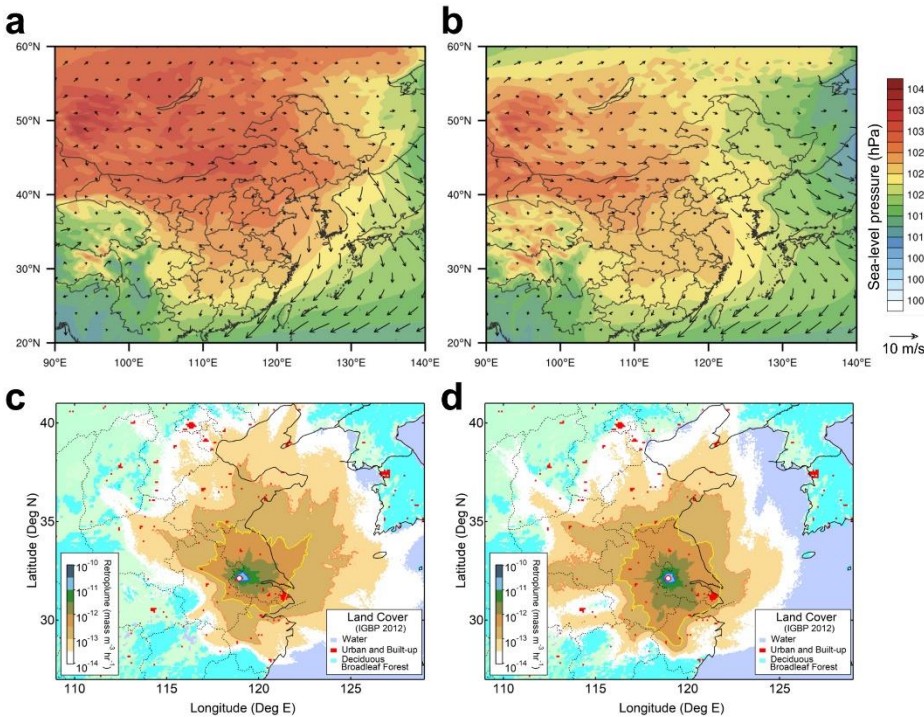

**Figure 9.** Averaged sea-level pressure and wind flows for **(a)** November 2011-January 2012 and **(b)** November 2013-January 2014, and **(c)** and **(d)** averaged retroplume from 2-day backward Lagrangian dispersion modeling for the corresponding period in **(a)** and **(b)**, respectively.