# Peer review of "Significant reduction of PM2.5 in eastern China due to regional-scale emission control: Evidences from the SORPES station, 2011-2018"

_Atmospheric Chemistry and Physics, 2019_

## Referee Comment (RC1) · Anonymous Referee #2 · 1 Jun 2019

The manuscript by Ding et al. investigated the roles of emission control and meteorology in decreasing PM2.5 in eastern China using long-term measurements. The results showed that control of open biomass burning and fossil-fuel combustion are the two major factors in reducing PM2.5 in early summer and winter, respectively. Such long-term measurements are very limited in China, which makes this study be important to understand the impact of emission sources, chemical mechanisms and meteorology processes on the reductions of aerosol species. This manuscript is overall well written, and I recommend it for publication in ACP.

Major comment:

[Figure]

High concentrations of PM2.5 in Yangtze River Delta were often associated with the transport from north China, particularly in winter season. Considering that the air quality in Beijing and Hebei province has been significantly improved during the last 6 years, I suggest the authors expanding some discussions how air quality improvement in eastern China was potentially associated with that in northern China.

---

## Referee Comment (RC2) · Anonymous Referee #3 · 6 Jun 2019

The manuscript by Ding et al., reported a long-term continuous trend of PM2.5, chemical species, and the precursors at the SORPES station in Nanjing, which is defined as a regional background station in the YRD region. With application of LPDM and comprehensive analysis with other supporting data, the authors investigated the impacts of emissions from fossil fuel combustion and open biomass burning and of year-to-year meteorology on the trends of primary and secondary PM2.5 in this region. The study revealed the effect of air pollution control measures over the YRD region in the past years. The paper is structured and written in a clear, thorough, and objective fashion that gives readers a clear understanding of the trends of PM2.5, related compounds and the effect of air pollution control measures. I recommend the paper to be published

after minor revisions. Detailed comments are as follows:

1. Page 4 Line23, Fig. 2 shows the trends of PM2.5 mass concentration and the two key precursors (SO2 and NO2) since 2011, and the main PM2.5 chemical components (BC, SO42- and NO3-) since 2013. Please also give the trends of NH4+ and NO.

2. Page 4, Line 32-34, "Among the two precursors, SO2 showed an even more significant reduction with an annual decrease about 17

3. Page 5, Line 1-2, "...achieved a very big success of air pollution prevention from desulfurization in power plant factories in recent years", this effect is not only due to the desulfurization in power plants, but also the measures including "replacement of coal with natural gas or electricity, etc".

4. Page 5 Line 13-14, "Here the results show that the efforts in reducing PM2.5 also co-benefited to the mitigation of global warming", please give more evidence or reference.

5. Page 5 Line 21-23, to examine the change before and after the "Ten measures", the authors separate the time period into 2011-2014 and 2015-2018. Since the "Ten measures" policy was released in August, 2013; why did not the authors separate the time period into 2011-2013 and 2014-2018?

6. Page 5 Line 3, please give the full name of the shortened "TRMM" when it is the first time to appear.

7. Page 5 Line 17, "Intensive emission from these activities could cause a secondary maximum of PM2.5 in early summer", K+ is a tracer of primary pollutants from biomass burning, therefore it should be primary instead of secondary.

8. Page 6 Line 14 and Line 25, "dominate" should be "dominant".

9. Page 7 Line 1, "due to efficient control from large elevated coal burning sources, such as power plants" and coal replacement with natural gas or electricity.

10. Page 7, Line 12-15, another reason is that the Ox concentrations, or the atmospheric oxidization capacity has been increasing in recently years, which will also enhance the formation of nitrate.

11. By only conducting the LPDM simulations based on the fixed MIX emission inventory for cold season (three months), could they quantify the influence of emission reduction and year-to-year change in meteorology?

12. Conclusions 2. an increased nitrate fraction in PM2.5 was observed because more NH3 were available for nitrate formation in the condition of reduced sulfate associated with a substantial reduction of SO2 and a moderate decrease of NOx. Another reason is the increase of Ox and atmospheric oxidization capacity.

13. Figure 4, why did the authors separate by 2012-2014 and 2016-2018 instead of 2012-2013, 2014-2018?

---

## Author Comment (AC1) · 20 Aug 2019

Response to Referee #2

The manuscript by Ding et al. investigated the roles of emission control and meteorology in decreasing PM2.5 in eastern China using long-term measurements. The results showed that control of open biomass burning and fossil-fuel combustion are the two major factors in reducing PM2.5 in early summer and winter, respectively. Such long-term measurements are very limited in China, which makes this study be important to understand the impact of emission sources, chemical mechanisms and meteorology processes on the reductions of aerosol species. This manuscript is overall well

written, and I recommend it for publication in ACP. Major comment: High concentrations of PM2.5 in Yangtze River Delta were often associated with the transport from north China, particularly in winter season. Considering that the air quality in Beijing and Hebei province has been significantly improved during the last 6 years, I suggest the authors expanding some discussions how air quality improvement in eastern China was potentially associated with that in northern China.

Response: Thanks a lot for the suggestion. We will conduct some additional results based on back-trajectory cluster analysis to evaluate potential reduction associated with emission reduction from different regions, including the northern China.

Response to Referee #3

The manuscript by Ding et al., reported a long-term continuous trend of PM2.5, chemical species, and the precursors at the SORPES station in Nanjing, which is defined as a regional background station in the YRD region. With application of LPDM and comprehensive analysis with other supporting data, the authors investigated the impacts of emissions from fossil fuel combustion and open biomass burning and of year-to-year meteorology on the trends of primary and secondary PM2.5 in this region. The study revealed the effect of air pollution control measures over the YRD region in the past years. The paper is structured and written in a clear, thorough, and objective fashion that gives readers a clear understanding of the trends of PM2.5, related compounds and the effect of air pollution control measures. I recommend the paper to be published after minor revisions. Detailed comments are as follows:

1. Page 4 Line23, Fig. 2 shows the trends of PM2.5 mass concentration and the two key precursors (SO2 and NO2) since 2011, and the main PM2.5 chemical components (BC, SO42- and NO3-) since 2013. Please also give the trends of NH4+ and NO.

Response: Thanks for the suggestions. WE will show the trends of NH4+ and NO in the revised version.
Interactive
comment

2. Page 4, Line 32-34, "Among the two precursors, SO2 showed an even more significant reduction with an annual decrease about 17

Response: Here we don't understand the point as the sentence is incomplete, but we will rewrite this sentence to avoid any misunderstanding in the revise version.

3. Page 5, Line 1-2, ". . .achieved a very big success of air pollution prevention from desulfurization in power plant factories in recent years", this effect is not only due to the desulfurization in power plants, but also the measures including "replacement of coal with natural gas or electricity, etc".

Response: Thanks. We will include this point in the revised version.

4. Page 5 Line 13-14, "Here the results show that the efforts in reducing PM2.5 also co-benefited to the mitigation of global warming", please give more evidence or reference.

Response: Here our point is in the context of black carbon as a well-known forcer for global warming. We will rewrite this sentence to include this message and relevant reference.

5. Page 5 Line 21-23, to examine the change before and after the "Ten measures", the authors separate the time period into 2011-2014 and 2015-2018. Since the "Ten measures" policy was released in August, 2013; why did not the authors separate the time period into 2011-2013 and 2014-2018?

Response: Thanks for the suggestion. We separate the period into two periods, the first 3 and last 4 years, to investigate the overall trend in the past 7 years but not only the "Ten-Measures" alone. Considering year-to-year difference in meteorology, it is inappropriate to compare average for periods with 2 years versus 5 years. Multi-year average for similar lengths could minimize the year-to-year variation in meteorology.

6. Page 5 Line 3, please give the full name of the shortened "TRMM" when it is the first time to appear.

Response: We will add the full name in the revised version.

7. Page 5 Line 17, "Intensive emission from these activities could cause a secondary maximum of PM2.5 in early summer", K+ is a tracer of primary pollutants from biomass burning, therefore it should be primary instead of secondary.

Response: Here the "secondary maximum" means a peak following the maximum in seasonal (monthly) variation. It is not "secondary aerosol".

8. Page 6 Line 14 and Line 25, "dominate" should be "dominant".

Response: Thanks. We will change it.

9. Page 7 Line 1, "due to efficient control from large elevated coal burning sources, such as power plants" and coal replacement with natural gas or electricity.

Response: Thanks. We will include this point.

10. Page 7, Line 12-15, another reason is that the Ox concentrations, or the atmospheric oxidization capacity has been increasing in recently years, which will also enhance the formation of nitrate.

Response: Thanks. We will include this suggestion.

11. By only conducting the LPDM simulations based on the fixed MIX emission inventory for cold season (three months), could they quantify the influence of emission reduction and year-to-year change in meteorology?

Response: Here we only conduct simulations for the three winter months NDJ just because the following reasons: 1) The concentrations in the three months are dominantly high from the seasonal pattern. The winter emission reduction rate could reflect the overall annual results and show stronger signal of reduction efforts. 2) Winter has less precipitation in this region. The LPDM simulations cannot well characterize wet-deposition and secondary formation of PM2.5. So these method has less uncertainty in winter. We will include these points in the revised version.

12. Conclusions 2. an increased nitrate fraction in PM2.5 was observed because more NH3 were available for nitrate formation in the condition of reduced sulfate associated with a substantial reduction of SO2 and a moderate decrease of NOx. Another reason is the increase of Ox and atmospheric oxidization capacity.

Response: Thanks. We will include this point.

13. Figure 4, why did the authors separate by 2012-2014 and 2016-2018 instead of 2012-2013, 2014-2018?

Response: Same as the Response to the No.5 comment.
* * *

---

## Author Response (AR1)

**Response to Referee #2**

The manuscript by Ding et al. investigated the roles of emission control and meteorology in decreasing PM2.5 in eastern China using long-term measurements. The results showed that control of open biomass burning and fossil-fuel combustion are the two major factors in reducing PM2.5 in early summer and winter, respectively. Such long-term measurements are very limited in China, which makes this study be important to understand the impact of emission sources, chemical mechanisms and meteorology processes on the reductions of aerosol species. This manuscript is overall well written, and I recommend it for publication in ACP.

Major comment:

High concentrations of $PM_{2.5}$ in Yangtze River Delta were often associated with the transport from north China, particularly in winter season. Considering that the air quality in Beijing and Hebei province has been significantly improved during the last 6 years, I suggest the authors expanding some discussions how air quality improvement in eastern China was potentially associated with that in northern China.

**Response:** Thanks a lot for the suggestion. We added some additional results based on back-trajectory cluster analysis to evaluate potential reduction associated with emission reduction from different regions, including the northern China. Please see P7, Lines 20-35 and Figure 7.

**Response to Referee #3**

The manuscript by Ding et al., reported a long-term continuous trend of $PM_{2.5}$, chemical species, and the precursors at the SORPES station in Nanjing, which is defined as a regional background station in the YRD region. With application of LPDM and comprehensive analysis with other supporting data, the authors investigated the impacts of emissions from fossil fuel combustion and open biomass burning and of year-to-year meteorology on the trends of primary and secondary $PM_{2.5}$ in this region. The study revealed the effect of air pollution control measures over the YRD region in the past years. The paper is structured and written in a clear, thorough, and objective fashion that gives readers a clear understanding of the trends of $PM_{2.5}$, related compounds and the effect of air pollution control measures. I recommend the paper to be published after minor revisions. Detailed comments are as follows:

1. Page 4 Line23, Fig. 2 shows the trends of $PM_{2.5}$ mass concentration and the two key precursors ($SO_2$ and $NO_2$) since 2011, and the main $PM_{2.5}$ chemical components (BC, $SO_4^{2-}$ and $NO_3^-$) since 2013. Please also give the trends of $NH_4^+$ and NO.

**Response:** Thanks for the suggestions. WE added the trends of $NH_4^+$ and NO in the revised version. Please see the updated Figure 2.

2. Page 4, Line 32-34, "Among the two precursors, SO2 showed an even more significant reduction with an annual decrease about 17

**Response:** Here we don't understand the point as the sentence is incomplete, but we rewrote this sentence to avoid any misunderstanding in the revise version. Please see P4, Line 35- P5, Line 1.

3. Page 5, Line 1-2, ". . .achieved a very big success of air pollution prevention from desulfurization in power plant factories in recent years", this effect is not only due to the desulfurization in power plants, but also the measures including "replacement of coal with natural gas or electricity, etc".

**Response:** Thanks. We included this point in the revised version.

4. Page 5 Line 13-14, "Here the results show that the efforts in reducing PM2.5 also co-benefited to the mitigation of global warming", please give more evidence or reference.

**Response:** Here our point is in the context of black carbon as a well-known forcer for global warming. We rewrote this sentence with references.

5. Page 5 Line 21-23, to examine the change before and after the "Ten measures", the authors separate the time period into 2011-2014 and 2015-2018. Since the "Ten measures" policy

was released in August, 2013; why did not the authors separate the time period into 2011-2013 and 2014-2018?

**Response:** Thanks for the suggestion. We separated the period into two periods, the first 3 and last 4 years, to investigate the overall trend in the past 7 years but not only the "Ten-Measures" alone. Considering year-to-year difference in meteorology, it is inappropriate to compare average for periods with 2 years versus 5 years. Multi-year average for similar lengths could minimize the year-to-year variation in meteorology.

6. Page 5 Line 3, please give the full name of the shortened "TRMM" when it is the first time to appear.

**Response:** We already defined it in the last paragraph of Session 2.2.

7. Page 5 Line 17, "Intensive emission from these activities could cause a secondary maximum of $PM_{2.5}$ in early summer", $K^+$ is a tracer of primary pollutants from biomass burning, therefore it should be primary instead of secondary.

**Response:** Here the "secondary maximum" is a typo. It is "second maximum, which means a peak following the maximum in seasonal (monthly) variation.

8. Page 6 Line 14 and Line 25, "dominate" should be "dominant".

**Response:** Thanks. We already corrected them.

9. Page 7 Line 1, "due to efficient control from large elevated coal burning sources, such as power plants" and coal replacement with natural gas or electricity.

**Response:** Thanks. We included this point in the revised version.

10. Page 7, Line 12-15, another reason is that the Ox concentrations, or the atmospheric oxidization capacity has been increasing in recently years, which will also enhance the formation of nitrate.

**Response:** Thanks. We already included this point in the revised version.

11. By only conducting the LPDM simulations based on the fixed MIX emission inventory for cold season (three months), could they quantify the influence of emission reduction and year-to-year change in meteorology?

**Response:** Here we only conduct simulations for the three winter months NDJ just because the following reasons:
1) The concentrations in the three months are dominantly high from the seasonal pattern. The winter emission reduction rate could reflect the overall annual results and show

2)  Winter has less precipitation in this region. The LPDM simulations cannot well characterize wet-deposition and secondary formation of $PM_{2.5}$. So, this method has less uncertainty in winter.

We included these points in the revised version. Please see P8, Line 5-9.

12. Conclusions 2. an increased nitrate fraction in PM2.5 was observed because more NH3 were available for nitrate formation in the condition of reduced sulfate associated with a substantial reduction of SO2 and a moderate decrease of NOx. Another reason is the increase of Ox and atmospheric oxidization capacity.

**Response:** Thanks. We included this point in the revised version.

13. Figure 4, why did the authors separate by 2012-2014 and 2016-2018 instead of 2012-2013, 2014-2018?

**Response:** Same as the Response to the No.5 comment.

[revised manuscript text omitted]